# Professional Athletes’ Well-Being: New Challenges in Post-COVID-19 Times

**DOI:** 10.3390/bs13100831

**Published:** 2023-10-11

**Authors:** Ran Assa, Abira Reizer

**Affiliations:** Department of Psychology and Behavioral Sciences, Ariel University, Ariel 40700, Israel; abirar@ariel.ac.il

**Keywords:** well-being, COVID-19, resilience, identity, psychological support

## Abstract

The COVID-19 period was characterized as a traumatic period throughout the world. During the pandemic, sports organizations had to adapt to government rules and social distancing measures frequently and faced a challenging and complex period in keeping their athletes on a normal routine. Many athletes faced uncertainty regarding their present and future competitive context and personal worries, similar to society during the pandemic. Consequently, adverse effects on the mental health and well-being of athletes were reported in individual and team sports. This review seeks to explore the perceived impact of COVID-19 on athletes’ well-being and future considerations. This review suggests professional athletes’ well-being should receive more attention and will be addressed in the future for the benefit of the athletes and not just in favor of performance. Moreover, the emphasis on evidence-based psychological support such as stress management and athletes’ well-being in a high-performance sport context should increase. The post-COVID-19 period highlights the importance of broadening athletic identity into a more holistic scope that includes life–performance balance and personal values outside the sporting context. Lastly, developing and fostering resilience is complex yet fundamental for systems, when considering athletes’ personal context and providing them with professional skills outside of their professional domain.

## 1. Introduction

COVID-19 is a global health and socioeconomic crisis that has affected millions of individuals and many organizations around the globe since it emerged in late 2019 [1]. The pandemic impacted many dimensions of life in general and has changed the way people operate as individuals and as communities; it has also changed the way organizations operate in the workplace [2]. The global crisis affected every aspect of life, including the sports industry, as elite sports teams and organizations suffered an immediate financial impact with losses [3]. Major local and international sporting events, including the Olympic Games in Tokyo and the European Football Championship, were canceled or postponed [4]. Moreover, grassroots-level, or small sports clubs often function under difficult conditions, being partly driven by volunteers, and had to face an unemployment emergency [5]. In the competitive prism, many professional and semi-professional leagues all over the world paused, and there was a serious threat of ending all competitions for an unknown amount of time [6]. In addition, lockdowns associated with attempting to arrest COVID-19’s spread caused the prevention of sporting events with spectators [7].

The pandemic immensely affected individuals practicing physical activities outdoors, which were banned or severely restricted because of enforced isolation and lockdowns [8]. Under these conditions, athletes were forced to change all forms of training and to be away from appropriate training conditions and communicating with coaches [9]. The result of that was lost in their new daily, weekly, monthly, and annual routines, affecting the mental and physical status of professional players all over the world [6]. The immediate effects of COVID-19 created uncertainty in the world of professional sports and among athletes. In the back of their minds, a new possibility was being taken into consideration.

For some athletes who trained and prepared for the Olympics, for example, the postponement of the competition meant retirement and an eternal loss of opportunity [6]. In comparison to injured athletes who got an opportunity they did not expect, for some it meant an existential crisis. Overall, and similarly to the general population, one of the major outcomes of the pandemic was the impaired well-being and mental health of elite athletes around the world [10,11].

The COVID-19 pandemic implicated the professional sports sector like many other sectors around the world, as described above. Our focus of interest is the athletes themselves, specifically the effect the pandemic had on their well-being. The pandemic created many challenges for sports organizations, such as adapting to isolation policies, hygiene measures, and using technology on a daily basis. We address this issue because, while it has been popular in recent years, it is now more relevant than ever because of the nature of the reality that the pandemic made us see clearly.

Following this line of research, the current review has two main goals. First, we will review the challenges that affected athletes during the COVID-19 pandemic and their relation to the existing challenges within professional sports. Second, we will try to focus on the lessons learned from the pandemic regarding professional athletes’ well-being and make some recommendations for the future. The main goal of this review is to highlight the importance of professional athletes’ well-being beyond the obvious interest in performance.

## 2. Method

This review has been conducted following the guidelines established by the Preferred Reporting Items for Systematic Reviews and Meta-Analyses (PRISMA) [12].

### 2.1. Protocol and Registration

Our protocol was drafted using the Preferred Reporting Items for Systematic Reviews and Meta-Analyses (PRISMA) protocols [12] and revised by both authors and an external expert in the field of research methods. The final protocol was not registered but was in accordance with the Open Science Framework [13].

### 2.2. Eligibility Criteria

Studies were included only if they mainly focused on the psychological aspects professional athletes have faced in specific relation to COVID-19; the studies were published during the years 2019–2022.

### 2.3. Search Strategy

Several databases were used for the literature review, including PubMed, Web of Science, and Google Scholar. A search using “COVID-19”, “athletes”, “mental health”, and “well-being” generated 580 results. A refined search with the addition of” professional athletes” generated 138 results. A review of these identified articles related to medical influences (i.e., sleep or eating disruptions) on youth, recreational athletes, and the general population, which were not relevant to our interest. These were reviewed for having a primary focus on the implication COVID-19 had on athletes’ well-being. This left 21 articles to be included in the analysis. An overview of these is provided in Table 1. Since the relevant papers were scarce, we increased our scope to include studies pertaining to critical changes that affect athletes’ lives, such as uncertainty, deselection, injuries, and athletic career development, in the discussion.

### 2.4. Synthesis of Results

We grouped the studies by the main themes they suggested, as we were trying to identify the unique ways individuals and organizations have faced COVID-19. We counted the number of studies included in the review that potentially met our inclusion criteria and noted how many studies had been missed by our search (i.e., before 2019 and not directly related to COVID-19), then we searched for an additional literature review to support the findings. Finally, we have summarized the overall challenges and suggested future recommendations.

## 3. Results

### 3.1. Athletes’ Challenges during and Post-COVID-19

This part of the review will describe and articulate the key challenges athletes faced during the pandemic and the lockdown periods. It should be noted that all the following themes regarding professional athletes’ well-being were relevant before COVID-19, which only increased and highlighted their importance. We will address the possible changes needed in post-COVID-19 times in order to face these challenges. Our focus is to review the factors that can increase athletes’ well-being in the future.

#### 3.1.1. Distress and Mental Health Difficulties

During the pandemic, the uncertainty regarding the cancellation of events and the revision of athletes’ contracts with sports clubs increased the damage to the psychological state and stressors of athletes [10,14]. When they were needed the most, athletes were isolated from professional staff, and there was insufficient use of technology to fill in the gap. Professional sports psychologists and practitioners reported higher demand for online psychological counseling and diagnosis of psychological disorders; among these were fear of being infected, anxiety, eating disorders, and family conflicts [15]. As elite athletes and resourceful others could afford to seek help, others were left without guidance and support [16]. Many of them found themselves inactive or without direction and suffered from substantive psychological stress. There is a similarity between times of crisis and critical transition moments, such as injury or retirement, regarding athletic well-being. The danger is substantial, especially for those lacking a clear goal for the future after years in a high-performance environment; a sense of denial, poor coping skills, along with social isolation could lead to mental illness or burnout [16].

In relation to dealing with psychological difficulties, the literature reveals that athletes experience mental health problems at similar rates to the general population (e.g., [17]). However, athletes face many stressors that the general population does not, such as personal (financial instability) or competitive ones (rigorous training schedules, injury, and deselection), which can lead to an increased risk of mental ill-health [18]. During the outbreak of COVID-19, countries applied social distancing policies, but in some nations, restrictions were not as strict [19]. This phenomenon became apparent on social media, where athletes watched how some of their competitors were able to train regularly, putting them at a disadvantage when there was ambiguity regarding competitions [16]. In addition, there is a stigma surrounding help-seeking behaviors for athletes who turn to professionals [20]. As some players admitted, they are reluctant to be honest when with the psychology practitioner because of feeling little value in disclosure or the fear of exposing their real feelings and the truth about themselves [20]. As some sports psychologists claimed at the very beginning of the pandemic, “Never has there been a more important moment for mental performance consultants to be accessible to their clients” ([16], p. 271). In other words, COVID-19 highlighted the attention needed regarding athletes’ well-being in general and specifically in times of crisis.

The following themes were found relevant to mental health and athletic well-being in the papers we reviewed. First, we will review each theme. We will provide suggestions for the specific use of each of them that could be implemented in sports settings in the recommendation section.

#### 3.1.2. Lack of Psychological Support

The first challenge COVID-19 emphasized is the need for psychological support. In recent years, organizations and sports clubs have made huge steps in understanding that need [18]. Despite that, the amount of time allocated for that purpose is insufficient, and, at times such as the COVID-19 crisis, athletes suffer. The main barrier to the lack of psychological support implementation is, unsurprisingly, a lack of financial resources [21]. A study that examined the application of sport psychology in English football clubs reported that time and staff resources also had an impact on the delivery of sport psychology [20].

Times such as the pandemic carry uncertainty and ambiguity, which are associated with increased negative effects and feelings [22]. These turbulent events cause physiological arousal such as stress and pressure on a daily basis [23]. Combined with the confinement that governments applied during the pandemic, stress can be exacerbated by the overall lack of relations with others that serve to maintain general well-being [24]. In an article that attempted to understand how the pandemic impacted Italian athletes, it was found that there was “a significant increase of stress and dysfunctional psychosocial states and a significant decrease of functional psychosocial states compared to situations prior to COVID-19” ([10], p. 86). These findings are in line with previous research on health crises in the general population, such as the 2003 SARS outbreak in Hong Kong, which was called a “mental health catastrophe” with long-term psychiatric morbidities [25].

Traditionally, psychological support services to athletes have been employed face-to-face in a conventional consultancy method [26]. Although these services are prevalent in the field of sports, the environment in which the area operates continues to evolve [27]. COVID-19 accelerated the interest in the use of technology in sports psychology services, together with the related lockdowns and social distance policies. In addition, it is important to highlight that athletes often train and live apart from their family, friends, and community; during COVID-19, that lack of social support was noticeable and decreased one of their resources for dealing with the crisis. These conditions (e.g., time, location, long working hours) are characterized as atypical in the professional sports industry, along with working in geographically dispersed teams [28]. If sports teams are separated by time and/or distance, sports psychologists should consider incorporating technology into their service. In times of crisis, there is growing demand from individuals and organizations to establish larger, and technologically integrated psychological support services.

#### 3.1.3. Athletic Identity

In their journey to become elite athletes, individuals inevitably encounter many critical moments that arouse anxiety and are associated with changes in their own identity [29]. Transitions during an athletic career are also an integral part of the life of an athlete, and they can involve selection or deselection, injury, or retirement. Some of the transitions could be predictable, but some are unpredictable or involuntary and can impact athletes’ mental health and well-being [30]. Research literature has established that individuals who attribute high importance to their sport possess a strong athletic identity and could face transitions in a more complicated way [30,31,32,33]. Athletic identity has been described as the degree to which an individual identifies with the role of an athlete [34]. Some athletes might even retire because of the difficulties, whether voluntary or involuntary. Similarly, COVID-19 has been a critical moment for athletes, especially for those who have spent a lot of resources and time on their sport and might have a very dominant athletic identity [11].

Having an athletic identity benefits elite athletes, as it is related to high levels of commitment; however, focusing solely on the athletic role could come at the expense of exploring other roles to identify with [34]. In contrast to exploring other roles as part of expanding one’s self-identity lies a process in which an individual restricts themselves to a single dimension. This is called identity foreclosure. This process relates to athletes’ belief that engaging with other identity roles could lead to negative effects on performance [35]. Hence, during a critical time such as injury, retirement, or COVID-19, individuals with a strong athletic identity could be unable to engage in that role because of restrictions and the cancelation of competitions. The loss of a dominant part of their identity could lead them to experience emotional disturbance, uncertainty, and disorientation, and it could even cause them to decline due to mental health issues [31]. The pandemic confronted athletes with another critical moment where the inability to engage with their dominant area of life and identity could invoke emotional responses and cause a negative outcome to their well-being.COVID-19 can be considered traumatic for many athletes around the world, similar to any transition during the course of an athletic career; therefore, organizations and stakeholders should consider the importance of professional support for individuals’ identities as much as any other prevalent transition. There are similarities between specific critical moments, such as an injury, to some of the lockdown experiences. A common feeling that athletes experience while injured and out of their sport is a lack of control [11]. As a result of their lack of access to training facilities and maintaining their fitness, athletes’ lack of control could lead to a decreased sense of autonomy [36]. Their perception of recovery, which is associated with a sense of control and autonomy, can impact how quickly they return to their sport [36]. Similarly to being injured, social constraints may cause a sense of alienation from teammates, coaches, and even themselves [37].

#### 3.1.4. Individual and Team Resilience

Recent research regarding the effects of the COVID-19 pandemic on the sports industry and athletes has highlighted several consequences of this unprecedented event. Among them were financial insecurity, changes to training, and reduced access to facilities and equipment [38]; increases in perceived stress and dysfunctional emotions [10] decreased motivation and concerns about future performance [39] and feeling a loss of access to one’s sport, support, and identity as an athlete [40]. These effects on athletes point to a need to better understand these difficult experiences in the context of global adversity [41]. A suitable psychological construct for understanding these events is resilience; this can be used as a framework that is often associated with performance success and positive mental health outcomes [42].

Psychological resilience has drawn immense interest from researchers and organizations over the past decades in order to better understand its meaning and nature (e.g., [43,44]). Over the years, the challenge of determining whether resilience is a trait, a process, or an outcome has been apparent. In the last decade, a shift toward a process-focused approach has been taken as the term “resilience” needs to be considered as an individual’s interaction with their environment [40]. The prevalent definition of the construct is the positive adaptation of adversity, although it was highlighted that in addition to adversity, which is often perceived as negative, resilience is also needed in a positive situation [26]. In a preliminary study, they defined resilience as “the role of mental processes and behavior in promoting personal assets and protecting an individual from the potential negative effect of stressors” ([45], p. 675). They added that this definition takes into consideration the conceptualization of resilience as both a trait and a process, the context of one’s environment, and the fact that it is developed over time [44]. In terms of dealing with adversity, COVID-19 could be seen as a time that affected athletes and forced them to react in a positive or negative manner.

COVID-19 and the consequent national lockdowns could be conceptualized as times of adversity since they were moments of hardship and suffering, and the concept of adversity is associated with trauma, difficulty, distress, or a tragic event [46]. This conceptualization also fits with the theoretical position that views COVID-19 as a critical event in a sporting career where changes can occur along with negative consequences and immense uncertainty. There is limited empirical evidence to rely on when trying to understand resilience as it was displayed in relation to COVID-19. In addition, the pandemic was an unpredicted barrier to athletes, and therefore neither they nor the general population could be prepared with proper resources and strategies [47]. However, an interesting model of uncertainty distress fits into that type of situation: A way to describe the impact of adversities and adverse situations that are simultaneous [48]. Dominant proactive factors of resilience, including perceived social support, a sense of meaning, and a motivational climate, were disrupted due to the crisis created during the pandemic. Athletes either did not have sufficient resources or could not access them at other times [40]. Contrary to past crisis events in which community members joined together, physically and socially, with a common purpose and energy to help each other, one of the most notable effects of the pandemic was social isolation. Forcing people to be apart also impacted individuals’ ability to remain resilient as social support and working together as a community diminished as resources [49]. Challenging as it was, athletes still engaged in their resilience process prior to and through the threat of COVID-19 to their physical and mental health.

#### 3.1.5. Summary

In sum, this section presented the challenges that COVID-19 created or highlighted regarding the sports industry and athletes’ lives around the world. As a result of the immediate cancelation of sports competitions around the globe, athletes lost their access to facilities and coaches and needed to be isolated, similar to the general population. They also experienced psychological difficulties such as anxiety, stress, and fears about their financial and professional future. In addition, other impacts, including a threat to their athletic identity and a lack of resilience resources, were apparent.

## 4. Discussion

Although the COVID-19 pandemic’s effects around the world have decreased in comparison to the beginning of the pandemic, it will continue to push organizations and individuals into necessary changes in the way we live our lives. In a sporting context, it is important to note that although it was an unprecedented event with monumental effects on athletes and sports fans, it had similar characteristics to other researched topics. Thus, it needs to be considered and treated in a similar way to previous research.

In critical moments such as those described above, along with the negative outcomes, there are opportunities for a positive future. Difficult moments have the potential to provide a chance to improve self-knowledge, question values and beliefs, and thrive psychologically. Taking responsibility and engaging in self-exploration can assist in growth after traumatic or adverse events [50,51].

Our aim is to address the ways sports organizations could use this opportunity to learn from the past in order to build a better future for the sake of athletes. The following paragraphs will articulate and propose specific actions that could be applied by professionals and, last but not least, that could be considered vital by stakeholders.

The first term that was mentioned above is critical moments, which are “those frequently experienced moments in our lives where we must confront the anxiety associated with an important change in our identity” ([29], p. 25); see also [50]. COVID-19 could be considered critical a moment as a career or injury [11]. Critical moments could be small or large and have negative or also positive consequences; self-exploration and responsibility could lead to psychological growth [50]. Among the positive outcomes of critical moments are increased motivation that impacts one’s appreciation of their sport, the development of non-sports-related areas of the self, and enhanced “mental toughness” [37]. In other words, critical moments provide the opportunity to develop, and athletes must learn from and seize them as they often face difficult moments that are similar in ways to COVID-19.

## 5. Future Recommendations

Our recommendations concern (a) professional sports organizations ranging from youth to senior individual and team athletes, (b) sports psychologists and practitioners who work with professional athletes, and (c) professional athletes, coaches, and other relevant staff members.

We recommend that there is a need to expand athletes’ psychological support in any way possible. Sports psychology has become increasingly recognized and accepted within many sports domains over the past decade (e.g., [52]). In fact, understanding the role of psychological skills as fundamental for success and further development is increasing in organizations [53]. This is not surprising because of the understanding research has established regarding athletes’ experiences. Moreover, expert athletes probably invest more in dealing with adversities, as they experience a higher intensity of idiosyncratic emotions and somatic symptoms; thus, they can better cope with stressful and uncertain situations [10]. This does not mean, however, that psychological support in sports organizations is applied to its best at the moment. This review suggests organizations and stakeholders increase the amount of psychological support given to athletes.

We should recognize the potential aids for that cause as it cannot be achieved by psychologists or practitioners alone. Perhaps highlighting the role of the coach and staff members as partners in increasing psychological support is necessary. A fundamental step is to increase the level of psychological understanding among coaches, as it appears to be low at present [21]. Coaches and staff members are integral parts of facilitating psychological interventions and can even be used as sensors for detecting a call for help, as they spend a considerable amount of time with athletes [54]. Since coaches have a great influence on athletes, they are currently at risk of providing suboptimal psychological support, and this is even more important for an athlete at an early age. Thus, practitioners should strive to influence coaches’ education regarding their role in promoting psychological support; this could be accomplished by delivering content, including the importance of delivering adequate psychological advice to athletes [21].

The other impact that coaches could facilitate is diminishing the stigma of help-seeking behaviors. It is evident that there still is a stigma attached to sports psychology that causes resistance from some coaches and athletes [55]. Therefore, helping to change that notion would be important to provide coaches and staff with education about sports psychology and the ways it can benefit performance and well-being. It should be noted that the sport psychology service is just as concerned with performance as it is with well-being. However, educating coaches about the importance of a holistic approach and highlighting the role of well-being in athletic performance is not sufficient. Increased guidance regarding the emotional and cognitive development of athletes would help deliver psychological services and meet the needs of the players [21].

Future work might expand our understanding of the predictors, moderators, and mediators that build athletes’ resilience and well-being. For example, recent research has found that autonomous motivation was related to athletes’ capacity to cope with the pandemic in a resilient way [56]. Future work could also provide a more solid theoretical model and develop interventions for supporting athletes’ well-being in uncertain and troubled times. Finally, practitioners could provide more one-to-one support along with on-field or education sessions that genuinely support youth players within and beyond sports environments.

The second part of increasing psychological support is concerned with the use of new media and technology in this area. The COVID-19 crisis has increased the importance of athletes’ need for accessible sports psychologists and practitioners. For self-employed practitioners who do not work within sports organizations, there has been a shift in practice toward online services [11]. The advantages of online services are many: they are portable, less expensive, and can be delivered to athletes with no geographical boundaries. This means that psychological support could be more accessible as practitioners are often challenged by time and distance constraints [57]. During the COVID-19 lockdowns, many Italian elite athletes reported using the internet to contact coaches and professionals in order to ensure their physical and psychological health [10]. The use of online platforms in the post-COVID-19 period should be increased and incorporated within sports organizations. Since working with elite athletes includes long travels, vast geographical distances, and different time zones, it is only natural that psychological support could be provided through online means. We suggest that practitioners present online workshops and educational presentations using online platforms, especially to young athletes who are familiar with this technology. When using an online session, athletes can choose their convenient hours before or after training and while on a summer vacation in a way that will not disrupt the therapeutic alliance continuity. Moreover, the gamification of any of the relevant subjects within sport psychology could be more useful than face-to-face content.

### 5.1. Identity: Broadening Identity/Being More Than an Athlete

The COVID-19 pandemic disrupted athletes’ access to sports facilities, paused their preparations for competitions, and impacted their daily lives immensely. For some, it meant that they would not be able to compete again and were forced to retire. For those individuals, considering and planning for future paths was needed, as a lack of ability to plan or prepare for life without sports is associated with negative emotions [32]. The impact of the pandemic could potentially suggest only negative outcomes, but it could also be seen, as mentioned above, as potential for self-growth. There is established research regarding athletic identity and the relation between athletes’ performance identity and well-being [47]. Some athletes discovered the importance of expanding their identity beyond sport, as it was no longer feasible for them to connect with their role in sport [41]. The need for athletes to have multiple narratives to story their lives has also been highlighted, as focusing solely on a performance or athletic narrative may be problematic when it does not unfold as expected [58]. Unfortunately, this was the case for many athletes during the pandemic. In contrast, athletes who could create an alternative narrative for themselves during a crisis or when not engaging with sports would be empowered and develop.

In a study that interviewed Canadian athletes who were preparing for the Tokyo 2020 Olympic Games, one athlete revealed that cultivating a life outside of sport during the pandemic helped her achieve her athletic goals and deal with multiple stressors [41]. In addition, the importance of being more than an athlete and expanding one’s identity could lead athletes to the development and maintenance of resilience [59]. It is important to understand that young athletes in their adolescence should engage in developmental tasks, including exploring and questioning their identity [60]. Despite that, many young athletes construct their lives mainly within a sports context [31]. Taking into consideration the fact that the statistical chances of succeeding as a professional are low, they could have an identity foreclosure that will impact their well-being [60].

Thus, we suggest that with respect to increasing psychological support and resilience, partitioners and psychologists should consider focusing on broadening their identity with their clients. There are some current models that could assist them in doing so. As crises such as the COVID-19 pandemic could represent a disruption to athletes’ careers, useful approaches could be, for example, a *whole-person* approach [61]. This is used to frame an athlete aspiring to participate and compete in sports and at the Olympic Games and who also has various non-sports pursuits [47]. Another approach is the *whole-career* perspective, which means “helping the athletes to bridge their past experiences, current issues, and anticipated future” ([47], p. 95); see also [62]. The crisis concerning transitions in an athletic career consists of the interplay between transition demands, internal and external resources and barriers, and coping strategies [63,64]. Self-identity could serve as a resource where, in critical times such as injury or retirement, parts of it that do not necessarily engage with the sport will be available for the individual.

It is worth mentioning that athletic identity is often dominant and has a central role as an internal resource for athletes striving for their goals [47]. In addition, athletic identity coexists and intersects with other identity dimensions, and the psychologist could assist the individual by focusing on the needed part for the relevant situation. Another theoretical yet applied approach regarding identity in sports that could be consulted is the existential one (e.g., [50,65]). The existential approach emphasizes meaning, values, responsibility, and situated freedom as fundamental constructs in human life [66]. In sports, this approach can help athletes clarify their challenges and identify sources of meaning and authentic goals and values [67]. This approach also aligns with the whole-person approach, which embraces personal growth. We suggest practitioners emphasize the use of broadening their athletes’ identity since we can see it has a positive influence on resilience and thus well-being.

### 5.2. Increasing Resilience: Individuals and Teams

The COVID-19 pandemic caused athletes many problems, both on and off the field. As mentioned above, individuals had to face adversity during that time, which could be viewed in a similar way to the continuous struggles they faced during their athletic careers. The post-COVID-19 period provides another opportunity to understand the role of psychological resilience in sports and to decide whether it could serve athletes’ well-being. In a recent paper, Sarkar and Page (2022) [68] suggested that it is essential to clarify the meaning of resilience and what it is not to create a common language and a consistent approach to resilience development. The foundation of resilience relies on personal qualities, also known as psychological factors, that protect the individual from negative consequences [45]. Among those qualities, there is a differentiation between personality characteristics (i.e., belief in oneself and one’s ability), psychological skills and processes (i.e., an awareness of oneself, others, and the environment), and desirable outcomes (i.e., regulating thoughts, mental images, and emotions) [68]. Thus, practitioners could facilitate the development of self-regulation with athletes, which could serve them as a protective factor of resilience [40]. Developing self-regulation, for example, would promote a sense of control and mastery, which can increase resilience [59]. Another applied strategy to increase resilience is to challenge individuals’ cognitive appraisals by drawing evidence from past experiences. Athletes who can identify and reflect on overcoming adversities, such as injury or dealing with the challenges of COVID-19, could be able to build a more resilient future [40].

There are a number of studies that have examined the way resilience can be developed in athletes. Fletcher and Sarkar (2016) [69] developed and piloted a “mental fortitude training” program for athletes that added a challenging mindset and a facilitative environment to their personal qualities. In a challenge mindset, athletes learn to change the way they look at stressors and shift them into growth opportunities. To improve this ability, athletes need to learn the more fundamental skill of identifying thoughts that are triggered by activating events and how they react to them [69]. To do so, individuals must be aware of their ABC: “an activating event (A), their beliefs (B) about the activating event, and the emotional and behavioral consequences (C) of those thoughts” ([68], p. 44). Recognizing and challenging counter-productive thoughts could help athletes be more informed about the way it makes them more vulnerable to the negative effects of stress [68]. In a facilitative environment, there is a focus on two fundamental notions to develop resilience: challenge and support [40]. The challenge involves the system’s (coaches, staff, athletes) expectations for accountability and responsibility, and support refers to enabling athletes to develop psychological characteristics and assists in increasing learning and building trust [68].

Much can be learned from an organizational point of view regarding developing resilience. Sarkar and Page (2020) [68] suggested that resilience could be developed with the assistance of coaches and supporting staff by using transformational and shared leadership. Moreover, creating an environment of learning and enjoyment, along with the creation of a team social identity, could facilitate resilience. Throughout the pandemic, the value of learning was magnified. When times of turmoil such as the pandemic disrupt the normal daily routine, there arises a tension between trying to maintain normalcy and changing everything that has served us so far [70]. Resilient teams, on the contrary, stick to what they already know and do and then try to reconfigure and redeploy, as the goal is to create order from chaos by leveraging current knowledge and experience [70]. Spending time and effort to reflect on experience is important to adapt and update learning. In addition, sharing information and learning from others rather than relying on trial and error can help to build collective resilience [70].

It is important to refer to mental health when mentioning resilience in a high-performance context. There is criticism regarding mental toughness in sports for its narrative stigma as athletes strive to be resilient and, if not, are in fear of being labeled as weak [71,72]. Resilience can be viewed as a spectrum as the important notion is to identify when it is functional regarding athletes’ adaptive purpose [41]. If we could conceptualize resilience as supportive of mental health, athletes would be better at recognizing when they are facing adversity and when it is not healthy or adaptable to keep pushing [73]. When promoting the development of resilience, it is critical to ensure it is accomplished in a way that promotes, rather than diminishes, mental health and supports athletes’ well-being. As this theme became highly relevant during COVID-19, literature and applied work regarding resilience have accelerated in the last couple of years [40,45,68]. We recommend that practitioners and sports organizations try and build individual and collective resilience using one of the ideas presented above or others. By doing so, they can help athletes be prepared to cope with future challenges and not affect their mental health negatively.

We recommend that in the near future, organizations should increase the amount of psychological support they provide, starting from academies and going up to the professional level. There has been an increase in the use of sports psychology within an athletic context in the past decade. However, it is still not sufficient and should be prioritized as a fundamental part of any sports organization. Athletes face many difficulties as part of their profession, as many of them struggle with uncertainty regarding their careers. The COVID-19 pandemic has shown us that our very existence is full of unpredicted events that affect our lives, and while some people can respond to ambiguity and thrive, others can suffer greatly. We know that there are differences in people’s ways of dealing with uncertainty, and this can lead to severe levels of distress. Increasing budgets, working with professionals, and trying to decrease the stigma about mental health are the very first steps that organizations can take in creating a larger support system for athletes. If there is a lesson to be learned, stakeholders should facilitate individuals’ ability to cope and deal with adversities, difficult emotions, and uncertainty as the world is changing rapidly. One example of a useful program to implement with youth players is the Elite Player Performance Plan (EPPP) within English soccer academies, which aims to integrate sport psychology support regularly as part of athletes’ development process. The provision of this kind of program within sports organizations will allow a better application of psychological support. The ultimate goal is for organizations to consider psychological support and athletes’ well-being as being as fundamental and critical as their performance.

In addition, supporting athletes’ mental health and well-being means trying to help them connect and explore other parts of their lives other than sports. That is why we suggest that understanding the whole of their identity is necessary. Identifying when it is best to focus on the athletic part to enhance performance is crucial for athletes. On the other hand, knowing when excessive contact with their athlete role can damage their performance and even well-being is also critical. There are interventions that should be implemented at an academy level due to the immense proportion of young athletes who simply cannot be professionals. To prepare for life after their athletic career is over or to deal with an injury and spend a period away from the sport, athletes need to be better equipped with tools to exist when the spotlight is off them. We have seen that interventions that support self-exploration or finding values and needs other than those related to sports can be successful in increasing well-being.

Finally, we recommend that in order to support athletes’ well-being, we need to focus on the way they handle adversity and negative emotions and increase their resilience. This subject has become very popular in sports psychology in the past decades, and research results are starting to appear in the literature (e.g., [68,69]). The post-COVID-19 period, as we suggested earlier, could be focused on implementing resilience models and methods in order to help athletes thrive and increase their well-being. There is not much work related to resilience interventions’ usefulness regarding athletes’ well-being, but the theoretical ground is set. Programs such as the “mental fortitude training” [69] that assisted British athletes during two Olympic Games are an example of what can be implemented to help them deal with adversity. It is important to note, however, that resilience development per se, without other significant factors, could undermine well-being [73]. In fact, because the term resilience can be interpreted differently, it is not naturally bound to well-being. Thus, we suggest resilience is used along with expanding athletic identity and increasing psychological support to serve the main goal—supporting athletes’ well-being—as part of a more holistic approach.

It is important to indicate that we did not address a key aspect in the current review regarding gender differences. Up until now, the majority of mental health research within the realm of elite athletes has been primarily focused on male athletes at the professional level [17,74]. The bulk of research pertaining to elite female athletes has been concentrated on collegiate-level student-athletes [18]. The available limited research that does encompass female athletes has revealed that females tend to encounter higher rates of mental health challenges compared to their male counterparts [17]. Only a few papers recently have discussed the stressors that are unique to elite female athletes in Italy and the UK [10,18]. Female athletes exhibited higher perceived stress scores in comparison to male athletes [10]. It is apparent that exploring barriers to accessing and/or utilizing psychological support within clubs, and psychological services that are being offered outside of the clubs is needed [18]. We think that further investigation is needed to address the recommendations concerning female well-being adequately.

Our overall conclusion refers to the main themes that we identified while conducting the current review and aimed to increase professional athletes’ well-being. The suggestion may broaden the psychological understanding of coaches and other supporting staff members and be considered when trying to build a team or individual resilience. Sport psychology practitioners could benefit from the recommendations above to plan an intervention or consider when identifying struggles associated with uncertainty, stress, or athletic identity individually and collectively.

This review has focused purposefully on the influences of the pandemic crisis on professional well-being. The current study expands this line of research by suggesting a wide-scope review of employees’ well-being, challenges, and opportunities in post-COVID-19 times. It aims to set the stage for further discussion of various ways in which research and practice can address the impacts of post-COVID-19 changes on athletes’ well-being. Indeed, many individuals suffer from impaired well-being and mental health problems and may experience COVID-19 as a devastating and long-lasting crisis [75,76]. However, others may find that COVID-19 was a turning point for changing their life path and for personal growth [77]. Due to the importance of the topic; professional sports organizations ranging from youth to senior individual and team athletes, sports psychologists and practitioners who work with professional athletes; and professional athletes, coaches, and other relevant staff members may benefit from the accumulative knowledge. We recommend that both public and private sports organizations enable athletes at all levels, from youth to senior professionals, to access online psychological support, whether a specific sport psychology practitioner who works solely in the organization or other freelancers. They can fully or partially fund an online meeting with sports psychology practitioners. In addition, we offer the opportunity for sports organizations to recognize the benefits when focusing on athletes’ well-being and not solely on improving performance when considering the psychological support delivered in their club. Building on our recommendation to increase mental health and athletes’ well-being by focusing and broadening their athletic identity or discussing potential ways to increase individual and team resilience could facilitate the responses athletes will face in the future in case of a dramatic event such as COVID-19 or a similar event with uncertainty.

## 6. Conclusions

In conclusion, the COVID-19 pandemic highlighted the importance of supporting well-being in the general population and especially among athletes, who were forced to pause their daily routines and encountered distress, psychological difficulties, and a lack of social support. With no competition or regular training, athletes struggled to stay in contact with their athletic identity, and some suffered because of that. Thus, we posit that there needs to be a shift regarding athletes’ well-being in sports organizations in the post-COVID-19 period.

## Figures and Tables

**Table 1 behavsci-13-00831-t001:** Literature Review Summary.

Classification	Sub-Classification	No
Journal	Psychology	6
	Sport and Exercise Psychology	5
	Sports	2
	Health and medical	2
	Others	3
Year	2020	9
	2021	5
	2022	7
Type of population	Elite Athletes	15
	Novice/youth athletes	3
	Recreational	3
	General Population	4

## Data Availability

No new data were created.

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
