# Peer review of "Professional Athletes’ Well-Being: New Challenges in Post-COVID-19 Times"

_behavsci, 2023, doi:10.3390/bs13100831_

Round 1

Reviewer 1 Report

I am glad that authors decided to discuss this relevant topic, since athletes were one of the most impacted group by Covid-19, by various factors. It is also important that the paper discusses practical resources to cope with this kind of global stressful situation. However, the relevance of the suggestions presented, would have much relevance if confirmed by empirical research(es).

Author Response

the author did not have any points or comments to respond

Reviewer 2 Report

This paper reviews a good number of previously published papers that address the issue of the impact of the Covid-19 pandemic on the psychological, physical and social well-being of athletes. The authors make recommendations at the end of the paper to address these issues.

The paper is well written and easy to follow. The recommendations are really just replications of statements contained in the literature that is cited. As such, they are quite general and not very original. I have three suggestions to improve the paper to make it more focused and usable to practitioners.

1. The distinction between amateur and professional athletes is quite muddy. I would think that amateur athletes have far fewer psychological and financial supports than professional athletes. Many professional athletes have their own nutritionists, trainers, etc. to assist them and their standard of living is much higher, particularly at the highest league levels. Their professional clubs provide more supports than amateurs receive, who mostly rely on government supports and their own finances. It would be useful to include articles that make the distinction between the professional and amateur athlete. I don't see how professional athletes at the highest levels had to face more stresses during the pandemic than prior to the pandemic. These seems to be stresses that they always face due to the competitive nature of their profession.

2. The authors do not distinguish between male and female athletes. That is unfortunate since there is much recent research that suggests that females suffered many more stresses during the pandemic than males in the general population. Female athletes have far fewer professional opportunities and thus have less access to psychological and financial supports. There might be a scholarly literature on this issue - I am not sure. But it is well worth pursuing. If male and female athletes face different stresses, they could require different recommendations that are more detailed than the general recommendations the authors provide.

3. The authors do not make it clear who the audience is for the recommendations. Are they speaking to amateur organizations (IAAF, etc.), professional organizations (FIFA, NFL, NBA, etc.), individual clubs (amateur or professional), governments or the athletes themselves? They should make more detailed recommendations, such as creating new administrative positions at the club or organizational levels, new types of therapies, new counselling strategies, player development programs, etc. This would increase the value of the paper and help determine who the audience is. How can the recommendations be acted on?

Author Response

21 August, 2023

Reviewers' feedback letter

Attached is a revised manuscript of the paper " Athletes’ well-being: new challenges in post-covid-19 times". Following comments from the associate editor and the reviewers, we revised, proofread, and re-edited our manuscript. We want to express our appreciation for the reviewers’ encouraging constructive, and helpful feedback. Their expertise and thoughtful feedback have immensely contributed to refining the quality and depth of my research. The constructive comments provided by the reviewers have been instrumental in enhancing the clarity and coherence of my paper. we will address each reviewer's comments and questions separately further on. We hope the revised version will be deemed suitable for publication in the journal Behavioral Sciences.

Reviewer 2 comments

This paper reviews a good number of previously published papers that address the issue of the impact of the Covid-19 pandemic on the psychological, physical, and social well-being of athletes. The authors make recommendations at the end of the paper to address these issues.

We thank the reviewer for the supportive comment

Point 1. The distinction between amateur and professional athletes is quite muddy. I would think that amateur athletes have far fewer psychological and financial supports than professional athletes. Many professional athletes have their own nutritionists, trainers, etc. to assist them and their standard of living is much higher, particularly at the highest league levels. Their professional clubs provide more supports than amateurs receive, who mostly rely on government supports and their own finances. It would be useful to include articles that make the distinction between the professional and amateur athlete. I don't see how professional athletes at the highest levels had to face more stresses during the pandemic than prior to the pandemic. These seems to be stresses that they always face due to the competitive nature of their profession.

Response 1. We agree with the reviewer's claim that amateur athletes would benefit from more finances and that a distinction between amateur and professional sports is needed. However, we would like to clarify that the purpose of our review is to focus on professional athletes' well-being. Previous work indicated that professional athletes did suffer from the pandemic and their professional career was severely restricted or damaged (see for example Rubio et al., 2021; Taku & Arai, 2020; Di Fronso et al., 2022; Whitcomb-Khan et al., 2021) Lockdowns, the restriction of social gatherings and the cancellation of professional competitions significantly affected the professional career of these athletes. However, although amateur athletes, may be affected in terms of impact to their mental health as many others around the world we think that the effects are different than professional athletes. Lack of Physical activity during covid-19 among the general population and mental health implications was addressed (e.g., Ai, Yang, Lin, & Wan, 2021; Shanbehzadeh, Tavahomi, Zanjari, Ebrahimi-Takamjani, & Amiri-Arimi, 2021). In addition, there is a stigma in sports regarding psychological support and mental health in comparison to the general population we thought it would be beneficial to review this trending topic.  Therefore, in the current work, we focused on professional athletes in particular

Furthermore, professional athletes are considered employees and have different obligations and career pursuits than amateurs. As employee well-being has been significantly damaged during covid time (for review see Hamouche, 2021; Reizer, 2022; Rudolph et al., 2021) we decided to focus on the professional athlete population.

However, following the reviewer's comment we have now emphasized that the current review focus on professional athletes and reflects the fact that it is not apparent and added clarifications to the paper (p.6, R. 294-297). The stresses they have faced during the pandemic reflected an uncertainty that had consequences on well-being and therefore the urge to review future challenges and solutions. We hope that the comments we added to the paper clarify this idea.

Point 2.  The authors do not distinguish between male and female athletes. That is unfortunate since there is much recent research that suggests that females suffered many more stresses during the pandemic than males in the general population. Female athletes have far fewer professional opportunities and thus have less access to psychological and financial supports. There might be a scholarly literature on this issue - I am not sure. But it is well worth pursuing. If male and female athletes face different stresses, they could require different recommendations that are more detailed than the general recommendations the authors provide.

Response 2. Thank you for the comment. Indeed, there are many gender differences whether on and off the sporting arena, the psychological and physical demands etc. The second author published a paper recently that discussed the effects on mental health women suffered during COVID-19. It is important to state that the current literature review regarding the gender differences in professional and elite sports is very limited that could not provide us with a clear understanding of the phenomena. We highly recommend further investigating the differences. We believe that there are gender differences that we cannot address in the current review and call for future work that will focus solely on that issue. (See p 11, lines 534-548).

Point 3. The authors do not make it clear who the audience is for the recommendations. Are they speaking to amateur organizations (IAAF, etc.), professional organizations (FIFA, NFL, NBA, etc.), individual clubs (amateur or professional), governments or the athletes themselves? They should make more detailed recommendations, such as creating new administrative positions at the club or organizational levels, new types of therapies, new counselling strategies, player development programs, etc. This would increase the value of the paper and help determine who the audience is. How can the recommendations be acted on?

Response 3. We understand that perhaps our intention to the relevant audience is not clear enough and have taken that into consideration while addressing that in the current version of the review (p 11-12, lines. 549-579 ). We thank you for your comment as they assisted us in making the review clearer. If our new comments do not suit your intention in this or the above comments, we will try to do so. Further suggestions were added in order to clarify the review's purpose (p 9, lines 414-416; p 10, lines 479-485).

We thank the reviewer for the constructive feedback

We are truly grateful for the reviewers' dedication and commitment to the scholarly pursuit of knowledge, which has undoubtedly enriched the overall content and presentation of our work. Once again, we extend my heartfelt gratitude to the academic reviewer for their invaluable role in shaping the scholarly trajectory of our work.

Sincerely,

Ran Assa. (Corresponding Author).

Abira Reizer, Ph.D.

Reviewer 3 Report

Strengths:

The topic of the article is interesting, considering the fact that it refers to issues that have affected the citizens of the entire planet, therefore also the athletes.

Weaknesses:

Questions for authors:

1. Who is this study for?

2. The title is about athletes. In the text of the article, references are made to the situation of elite athletes. Perhaps the change should be made in the title.

3. What is the research question/What are the research questions or research hypothesis/hypotheses?

4. What are the materials used to prepare this study?

5. What research methods did the authors use?

6. To whom are the recommendations addressed?

7. References to technology are based on claims from articles published in 2000, 2004. Over two decades have passed since then, and technology has evolved enormously. It is hard to believe that in 2020 athletes did not have access to a phone, laptop, tablet, internet, to get in touch with colleagues, coaches, to search for information and videos for training even in isolation (r. 109-112).

8. R. 329-336 states: "The use of online platforms in the post-COVID-19 era should be increased and incorporated within sports organizations." What would be the purpose of their use in the post-COVID-19 era? Where? At home or at training? At what point in training should they be incorporated? With what frequency?

9. We recommend systematizing the ideas presented, in such a way that the reader can form a clear picture of them.

Author Response

21 August, 2023

Reviewers feedback letter

Attached is a revised manuscript of the paper " Athletes’ well-being: new challenges in post-covid-19 times". Following comments from the associate editor and the reviewers, we revised, proofread, and re-edited our manuscript. We want to express our appreciation for the reviewers’ encouraging constructive, and helpful feedback. Their expertise and thoughtful feedback have immensely contributed to refining the quality and depth of my research. The constructive comments provided by the reviewers have been instrumental in enhancing the clarity and coherence of my paper. we will address each reviewer's comments and questions separately further on. We hope the revised version will be deemed suitable for publication in the journal Behavioral Sciences.

Reviewer 3

Questions for authors:

Point 1. Who is this study for?

Response 1. We intend to refer our review to professional athletes, sport psychology practitioners and coaches, and staff members with sports clubs (p 6, lines 294-297). However, we are not sure we understand the reviewer’s request properly. We can clarify it if necessary.

Point 2. The title is about athletes. In the text of the article, references are made to the situation of elite athletes. Perhaps the change should be made in the title.

Response 2. Thank you for the comment, we have changed the title and hope it makes it clearer to see that we refer to professional athletes.

Point 3. What is the research question/What are the research questions or research hypothesis/hypotheses?

Response 3. We have now clarified that this paper is a review that focused on the effect of the COVID-19 pandemic on professional athletes' well-being while mapping the potential predictors of professional athletes' well-being during the pandemic and research, we addressed that in the current paper (p 2, lines 70-83).

Point 4. What are the materials used to prepare this study?

Response 4. We have added in short the papers we have used to conduct this review (p 2, lines 70-83).

Point 5. What research methods did the authors use?

Response 5. We have added in short the method of the current review (p 2, lines 70-83). We have read around 30 articles regarding the implication of covid-19 on sports as only 15 studies discussed athletes' mental health and well-being.

Point 6. To whom are the recommendations addressed?

Response 6. This review is relevant to professional sports organizations (individual and team athletes), sports psychologists and practitioners who work with professional athletes, and professional athletes, coaches, and other relevant staff members (p 6, lines 294-297).

Point 7. References to technology are based on claims from articles published in 2000, 2004. Over two decades have passed since then, and technology has evolved enormously. It is hard to believe that in 2020 athletes did not have access to a phone, laptop, tablet, internet, to get in touch with colleagues, coaches, to search for information and videos for training even in isolation (r. 109-112).

Response 7. Thank you for your comment, we have now deleted these references from the paper.

Point 8. R. 329-336 states: "The use of online platforms in the post-COVID-19 era should be increased and incorporated within sports organizations." What would be the purpose of their use in the post-COVID-19 era? Where? At home or at training? At what point in training should they be incorporated? With what frequency?

Response 8. We have tried to clarify our intention in this paragraph (p 7, lines 357-359) and hope that it is more comprehensive now.

Point 9. We recommend systematizing the ideas presented, in such a way that the reader can form a clear picture of them.

Response 9. Further suggestions were added in order to clarify the review's purpose (p 9, lines 414-416; p 10, lines 479-485).

If our new comments do not suit your intention in this or the above comments, we will make an effort to do so.

We are truly grateful for the reviewers' dedication and commitment to the scholarly pursuit of knowledge, which has undoubtedly enriched the overall content and presentation of our work. Once again, we extend my heartfelt gratitude to the academic reviewer for their invaluable role in shaping the scholarly trajectory of our work.

Sincerely,

Ran Assa. (Corresponding Author).

Abira Reizer, Ph.D.

Round 2

Reviewer 2 Report

One of my comments addressed the lack of distinction between amateur and professional athletes in the body of the paper. That is still not addressed in the revised copy. The author(s) merely added the word "Professional" in the title of the paper. The reader that forgets the title will not see the distinction in the body of the text or in the conclusions since there are no references to any professional leagues or associations.

Author Response

Dear reviewer,

We have applied further changes in our work to clarify the distinction of the population in the interest of this paper. we hope that the changes in our methodology will help to better understand your comment. We would appreciate hearing how we can improve our work.

Reviewer 3 Report

With all due respect to the work done, from my point of view it is not a scientific article. Regardless of the additions made, only some recommendations remain for sports clubs.

Author Response

Dear reviewer,

We have applied further changes in order to clarify the measures taken in this paper's methodology. Regarding our recommendations, we would appreciate it if you could provide further feedback on any changes we can apply to our work.